# Adiabatic versus non-adiabatic electron transfer at 2D electrode materials

Dan-Qing Liu[1,2], Minkyung Kang[1,3], David Perry[1], Chang-Hui Chen[1], Geoff West[4], Xue Xia[5], Shayantan Chaudhuri [1,6], Zachary P. L. Laker[5], Neil R. Wilson [5], Gabriel N. Meloni[1], Marko M. Melander [7✉], Reinhard J. Maurer [1✉] & Patrick R. Unwin [1✉]

2D electrode materials are often deployed on conductive supports for electrochemistry and there is a great need to understand fundamental electrochemical processes in this electrode configuration. Here, an integrated experimental-theoretical approach is used to resolve the key electronic interactions in outer-sphere electron transfer (OS-ET), a cornerstone elementary electrochemical reaction, at graphene as-grown on a copper electrode. Using scanning electrochemical cell microscopy, and co-located structural microscopy, the classical hexaamineruthenium (III/II) couple shows the ET kinetics trend: monolayer > bilayer > multilayer graphene. This trend is rationalized quantitatively through the development of rate theory, using the Schmickler-Newns-Anderson model Hamiltonian for ET, with the explicit incorporation of electrostatic interactions in the double layer, and parameterized using constant potential density functional theory calculations. The ET mechanism is predominantly adiabatic; the addition of subsequent graphene layers increases the contact potential, producing an increase in the effective barrier to ET at the electrode/electrolyte interface.

[1] Department of Chemistry, University of Warwick, Coventry CV4 7AL, UK. [2] School of Materials Science and Engineering, Zhejiang University, Hangzhou 310007, China. [3] Institute for Frontier Materials, Deakin University, Geelong, VIC 3217, Australia. [4] Warwick Manufacturing Group, University of Warwick, Coventry CV4 7AL, UK. [5] Department of Physics, University of Warwick, Coventry CV4 7AL, UK. [6] Centre for Doctoral Training in Diamond Science and Technology, University of Warwick, Coventry CV4 7AL, UK. [7] Department of Chemistry, Nanoscience Center, University of Jyväskylä, P.O. Box 35, (YN) FI-40014 Jyväskylä, Finland. ✉email: marko.m.melander@jyu.fi; r.maurer@warwick.ac.uk; p.r.unwin@warwick.ac.uk

Electrochemistry offers a unique possibility to modify both the thermodynamics and kinetics of redox reactions by changing the electrode potential. Much of our present theoretical understanding of electrochemical kinetics is based on rather rudimentary treatments and model Hamiltonians which describe the kinetics in terms of simple but physically and conceptually well-defined parameters. Extracting atomic-level insight from these models remains challenging, as the parameters are often treated merely as fitting parameters or obtained from first principles for simplified systems which cannot be addressed experimentally[1]. This situation prevails even for the simplest (textbook) case of outer-sphere electron-transfer (OS-ET), where the redox couple is typically at a distance of at least a solvent layer from the electrode surface[2]. Depending on the degree of electronic coupling between the redox couple and electrode, OS-ET is classified as adiabatic or non-adiabatic, and identifying the degree to which OS-ET lies towards either limit is of considerable fundamental interest[3]. Studies of OS-ET at electrode surfaces modified with self-assembled monolayers[4,5] or a thin insulating oxide layer[6], which separate the redox couple from the conductive electrode surface, naturally push OS-ET towards the non-adiabatic limit, as confirmed by experiment and theory in the past[4–9]. However, the degree of electronic coupling for OS-ET at more commonly used bare electrodes has been much more difficult to study with both theory or experiment. As a consequence, we lack convincing microscopic pictures for OS-ET[2], the most elementary electrochemical redox reaction.

In the adiabatic limit, the pre-exponential factor is independent of the electron tunneling probability between the electrode and the redox couple, and the OS-ET rate is predicted to become independent of the electrode material, provided the electronic interaction between the redox couple and electrode is sufficiently strong[10,11]. In contrast, in the non-adiabatic limit, the rate is proportional to the density of electronic states (DOS) near the Fermi level (moderated by the electronic coupling efficiency)[5,12–14]. As a consequence, experimental tests of adiabatic versus non-adiabatic theory have been made with different electrode materials, as a means of examining the DOS-dependence of ET kinetics[15–18]. An independence of ET rate constant on electrode material is considered to indicate an adiabatic reaction, whereas an electrode-material dependent rate constant, is argued to mean the reaction is non-adiabatic. Yet, reported DOS effects on ET kinetics are usually weak, with non-adiabacity deduced from rate constant ratios on different types of metal electrodes of <2 or <1.3[5,18]. A general experimental issue for all such measurements is that OS-ET reactions are typically fast and close to the diffusion-limit of the experimental techniques employed[15,18], making kinetic assignments difficult.

It is increasingly recognized that true understanding of complex electrochemical reactions at the atomic level can only be reached by combining detailed experiments, theory, and simulations[19,20]. The necessity to account for the solvent, interface electrostatics, and the potential that is applied to the electrode, have made it difficult for theory and atomistic modeling to be reconciled with electrochemical experiments in the past, hindering atomic-level insight on OS-ET[1]. This situation has only very recently started to change due to advances in electronic structure theory[21].

Metal-supported 2D electrode materials[22–28] provide an interesting testbed for ET theory, as electrochemical activity can be modulated and controlled via the electronic interaction of the metal back contact/2D material[24,26,27,29–33]. This is especially the case for metal-supported graphene, a configuration with growing applications in electrocatalysis[24], and for corrosion protection of metal surfaces by graphene[34,35]. Although many earlier electrochemical studies of graphene, both exfoliated and grown by

chemical vapor deposition (CVD), considered material transferred to a Si/SiO$_2$ surface[24,30,36–44], the study of graphene as-grown on Cu not only provides a back contact for electrochemistry[28], but negates the need for polymeric transfer to a substrate, which may significantly contaminate the graphene surface and the resulting electrochemistry[45].

As-grown graphene on Cu presents small islands of bilayer (BL) and multilayer graphene[46,47] on a contiguous monolayer (ML) of graphene/Cu. These different distinct structural motifs can be targeted directly with scanning electrochemical cell microscopy (SECCM), a technique that provides spatially-resolved measurements at different locations on an electrode surface[37,39,48–54], while the response of unwanted pinholes can be detected[55] and eliminated from the analysis. SECCM delivers reasonably high mass transport rates[56], so that fast ET kinetics can be measured. The further use of co-located Raman microscopy and field emission scanning electron microscopy (FE-SEM) allow unambiguous characterization of the sites of the electrochemical measurements in a correlative multi-microscopy approach[49,57]. The Cu substrate surface crystallography upon which the graphene sits, can also be mapped with electron backscatter diffraction (EBSD) to determine any influence on the electrochemical response[58–63].

Herein, we address the question of adiabatic vs. non-adiabatic OS-ET to provide a new perspective on electronic control of electrochemistry at Cu-supported graphene. We consider the ET kinetics of $[Ru(NH_3)_6]^{3+/2+}$, a classic example of OS-ET[6,64–68], which has been employed for studies of outer-sphere electrochemistry at graphene[39,69], and does not adsorb on graphene at a detectable level[70]. SECCM multi-microscopy reveals that the ET kinetics is in the order ML>BL>multilayer graphene on copper. To explain this trend, we develop a theoretical model based on the Schmickler–Newns–Anderson (SNA) model Hamiltonian for ET accounting explicitly for the electrostatic interactions[71–73] in the double layer. The SNA Hamiltonian is parametrized using constant potential density functional theory (DFT) and used to study the degree of (non-)adiabaticity of OS-ET from rate theory, allowing us to connect atomistic structure and potential-dependent properties with the OS-ET rate. Detailed analysis of data for ML and BL graphene indicates a predominantly adiabatic mechanism, where the addition of subsequent graphene layers increases the effective barrier, by partially screening the electrode potential. The methodology and combined experiment/theory/simulation analysis we outline should be generally applicable to many electrochemical processes particularly for metal-supported 2D materials.

## Results and discussion

### Electrochemical rate theory and the Schmickler–Newns–Anderson Hamiltonian.
We establish a general theoretical framework for the treatment of OS-ET, which is tailored to the specific case of graphene on copper. Within general (electro)chemical rate theory, the reaction rate constant is given as[74]

$$k(E) = \kappa(E) e^{-\Delta G^{\ddagger}(E)/k_b T} \qquad (1)$$

where $\kappa(E)$ is a potential-dependent prefactor which accounts for the attempt frequency in transition state theory (TST) and may include effects beyond TST such as non-adiabatic corrections, solvent dynamics, nuclear quantum effects, among other. $\Delta G^{\ddagger}(E)$ is the free energy barrier, which depends on the electrode potential, $E$. The Butler-Volmer equation is applied most often to analyze OS-ET[75–78], in which the change of the rate with potential is lumped into a single effective parameter – the symmetry factor, α. More physically motivated models are based on the seminal work of Marcus,[79,80] which has been extended to include contributions from e.g. both

inner- and outer-sphere components[81,82], adiabatic and non-adiabatic kinetics[13], nuclear quantum effects[79], and the manifold of electronic states of the electrode material[5,83], all of which can affect the prefactor and the barrier in a physically motivated manner[14,84].

To explicitly include key microscopic parameters in a single theoretical formulation, several extensions[10,13,83,85,86] to the Newns–Anderson model Hamiltonian have been developed for electrochemical charge transfer at metallic, semiconducting, and graphene-based electrodes[10]. Compared to the Butler–Volmer equation, the different terms entering the extended Marcus-like theories or the SNA theory can be addressed and computed separately to understand the fundamentals of electrochemical reaction kinetics. Typically, the SNA Hamiltonian consists of electronic properties of the pure redox couple, the electrode, and their electronic interactions ($H_{el}$), solvent energy ($H_{sol}$), and solvent-molecule interactions ($H_{int}$). In this work, to account for the varying electrostatic interactions between the electrode and the redox molecule, we apply an extended SNA Hamiltonian[73] with an additional electrostatic term ($H_\phi$). Then, the total Hamiltonian reads:

$$H = H_{el} + H_{sol} + H_{int} + H_\phi \quad (2)$$

as depicted in the schematic in Fig. 1b. This Hamiltonian contains information about both the prefactor and the barrier and can thus be used to address both adiabatic and non-adiabatic ET rates[13]. As detailed in the Supplementary Information Note 13, the central parameters and quantities entering the Hamiltonian are physically well-defined and enable separation of the contributions of the solvent reorganization energy, electronic structure or DOS of the electrode, and the electronic coupling elements ($V$). These quantities, in turn, determine both the prefactor and barrier in Eq. (1), and this facilitates dissection of the rate constant to obtain fundamental insight on the adiabaticity and reaction barrier:

$$\Delta G^{\ddagger} = H^{\ddagger}(d, q^{\ddagger}, E) - H(d, q^0, E) \quad (3)$$

$$k_{red} = \kappa \times e^{-\frac{\Delta G^{\ddagger}}{k_B T}} \quad (4)$$

where $d$ is the distance between the redox center and the electrode and $q$ is the reaction coordinate. In the adiabatic case, the transmission coefficient is $\kappa \approx \frac{k_B T}{h}$. In the non-adiabatic case, it can be approximated as: $\kappa \approx \frac{2\pi}{h}\frac{V^2}{\sqrt{4k_B T \lambda}}$ (for details, see Supplementary Note 13). As all quantities entering the SNA Hamiltonian can be obtained from first principles calculations[1], the electrochemical ET kinetics can be understood with physically and chemically well-defined parameters and systems. Herein, this philosophy is used to interpret the experimentally measured OS-ET kinetics of $[Ru(NH_3)_6]^{3+/2+}$ on graphene/copper electrodes. Experimental observations are analyzed by computing the SNA Hamiltonian parameters from constant potential DFT calculations of ML and BL graphene on Cu(111).

**Experimental electrochemical rate measurements: voltammetric SECCM.** Voltammetric SECCM[59] (Fig. 1a) is employed to obtain potentiodynamic movies of the $[Ru(NH_3)_6]^{3+/2+}$ ET process at graphene/copper working electrodes of defined character, deduced from co-located correlative microscopy (*vide infra*) and the results are compared to the OS-ET model (Fig. 1b). Dual channel theta pipet probes were used, containing an aqueous solution of 1 mM $[Ru(NH_3)_6]^{3+/2+}$ and 50 mM KCl supporting electrolyte, with an AgCl-coated Ag wire in each channel to act as a quasi-reference counter electrode (QRCE)[54,87]. All potentials are quoted against this electrode, which had a stable potential[88] of 56 mV vs. saturated calomel electrode (SCE). The advantage of the theta pipet probe is that the meniscus landing of the SECCM tip is sensed via the ion conductance current that flows across a bias between the 2 QRCEs and so the meniscus cell can be landed on any surface[48]. Moreover, the ion conductance current across the meniscus informs on the stability and reproducibility of meniscus contact from point to point[48,89] (*vide infra*).

A typical area (referred to as "area 1" herein) scanned by a theta pipet (diameter ~700 nm) is considered first. A cyclic voltammogram (CV) was recorded at each position in the array (which make the pixels of the resulting potentiodynamic movies, *vide infra*), starting from a potential of 0 V, with the potential swept linearly with time to −0.6 V, and then swept back to 0 V. These measurements can be presented as a potentiodynamic movie (Supplementary Movie 1) of electrochemical (surface) current as a function of potential at each position (pixel).

Area 1 was characterized with FE-SEM (Supplementary Fig. 1a), and revealed contrast in image intensity across the surface, which suggests there are regions with different numbers of graphene layers. Further characterization by Raman microscopy identified the number of graphene layers, from the 2D/G ratio and the full-width half maximum of the 2D band (Supplementary Fig. 2), as ML (labeled A) and BL (labeled B) regions. For the spectra of region A (Supplementary Fig. 2b,c), the low ratio of peak intensities $I_D/I_G$ (0.073) indicates high-quality graphene[90]. These data enabled the co-located potentiodynamic maps and movies to be correlated with the graphene character.

Current density maps taken from Supplementary Movie 1 are shown in Fig. 2 for a substrate working electrode potential of $E_{WE} = 0.00$ V (a) vs. Ag/AgCl QRCE (beginning of CV), (b) −0.33 V (around the half-wave potential region) and (c) −0.60 V (limiting current region). The white outline on each map demarks the boundary between ML and BL graphene, identified from the SEM image. At 0 V (a), most of the surface shows little electrochemical activity (close to zero current detected on the pA scale), but there are a few pixels with a significant positive current (marked with white rings), attributed to pinholes in the graphene layer that exposes the Cu foil to the electrolyte solution,

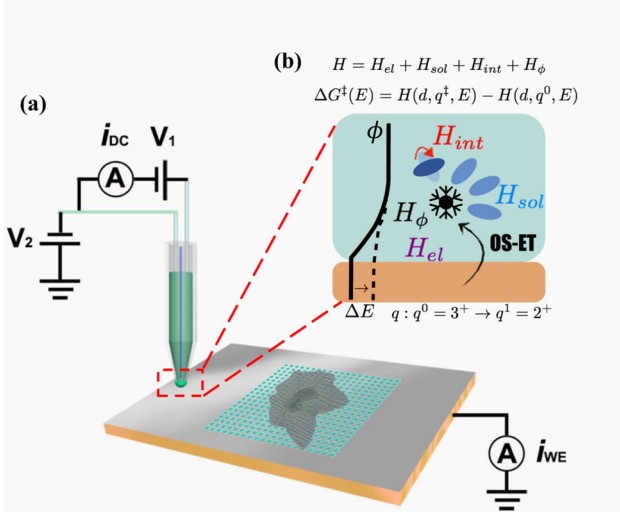

**Fig. 1 Integrated experimental and theoretical analysis of electrochemical outer sphere electron transfer (OS-ET).** Schematic illustrations of **a** the voltammetric scanning electrochemical cell microscopy setup with a hopping mode in which a dual channel probe was moved to and from the substrate working electrode surface (meniscus contact) at a series of predefined pixels and **b** the interactions included in the Schmickler–Newns–Anderson Hamiltonian and how they relate to the OS-ET barrier (see text for details).

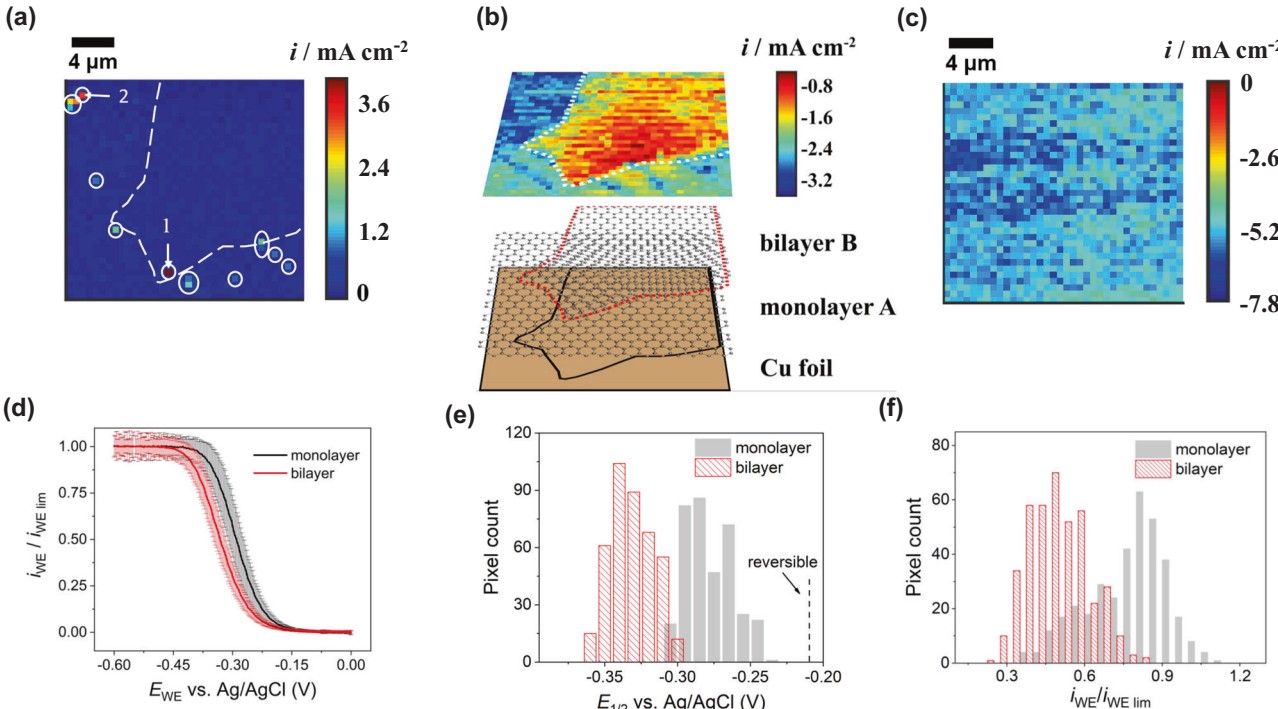

**Fig. 2 SECCM ET rate measurements of area 1.** SECCM images of area 1 for [Ru(NH₃)₆]³⁺ reduction at: **a** 0 V, **b** –0.33 V and **c** –0.60 V vs. Ag/AgCl quasi-reference counter electrode, extracted from potentiodynamic Supplementary Movie 1. The corresponding co-located FE-SEM and Raman image and analysis of this area are given in Supplementary Fig. 1a and 2b–d. **d** Average linear sweep voltammetry response for monolayer (ML) graphene (region A) from 355 measurements and bilayer (BL) graphene (region B) from 404 measurements (one standard deviation, s.d., of current). The scan rate was 0.5 V/s. **e** $E_{1/2}$ values measured on ML graphene ($N = 355$) and BL graphene ($N = 404$) of area 1 and **f** Histogram of $i_{WE}/i_{WE,lim}$ values at –0.33 V for ML ($N = 355$) and BL graphene ($N = 404$).

resulting in anodic dissolution in the chloride medium at this potential. CVs at these pinholes were extracted and examined, as exemplified by the responses in Supplementary Fig. 3; and further analysis to prove that the response is due to copper anodic dissolution is provided in Supplementary Note 4. A key attribute of SECCM is that these sites can be located, and then excluded from any subsequent analysis of graphene/copper electrochemistry. Note that any copper ions released at a pinhole in graphene are expected to be reduced at the same position due to the voltage scan range used, and there is no transference to complicate the response at subsequent pixels.

The extracted current density map at $E_{WE} = -0.33$ V (Fig. 2b), and the potentiodynamic movie at potentials in the kinetic region (Supplementary Movie 1), shows the electrochemical activity clearly correlates with the electrode structure. Notably, in the kinetic region (Fig. 2b), at a particular potential, the current recorded over BL graphene is lower than over ML graphene. At more cathodic potentials than $-0.5$ V, the steady-state current is diffusion-limited and more uniform over the entire surface (Fig. 2c).

The mean responses of $i_{WE}$ (normalized by the diffusion-limited current density at each pixel, $i_{WE,lim}$) vs. substrate working electrode potential, $E_{WE}$, for the ML and BL regions are shown in Fig. 2d. These data were extracted from regions that excluded pinholes and the boundary between the ML and BL regions. Evidently, the [Ru(NH₃)₆]³⁺/²⁺ redox process is faster at ML than BL graphene on copper, as also evident from the distributions of the half-wave potential, $E_{1/2}$ (Fig. 2e), the quartile potentials (Supplementary Fig. 5a,b), and the ratio $i/i_{WE,lim}$ at $-0.33$ V (Fig. 2f), where we analyze the ET kinetics (vide infra and see Supplementary Note 6 and 7).

We now consider data from a second area, area 2 (Fig. 3). The contrast of the FE-SEM image of this area (Supplementary Fig. 10b) is similar to that discussed for area 1, revealing the presence of ML and BL graphene, but there is also a small, even darker region, attributed to multilayer graphene. This was confirmed by Raman microscopy characterization of area 2 (Supplementary Note 2) that clearly identifies 3 types of graphene in different areas of the surface comprising ML (A), BL (B), and multilayer (C), where for the latter $3 \leq$ layer number $\leq 5$[91,92]. The corresponding SECCM potentiodynamic movie (Supplementary Movie 2) and a snapshot from the movie (current density) at a potential in the kinetic region, of $-0.33$ V vs. Ag/AgCl QRCE (Fig. 3a), shows again that there are significant variations in electrochemical activity that match to the local graphene character. The limiting current map in Fig. 3b ($E_{WE} = -0.60$ V) indicates that the measurements are reasonably consistent across the different graphene areas. These data were obtained with a pipet of diameter ~1 μm (Supplementary Note 8, Fig. 10a).

To compare the electrochemistry of ML, BL and multilayer graphene areas (marked as A, B and C in Fig. 3a), we plotted the $i_{WE}/i_{WE,lim}-E_{WE}$ responses for each, obtained from the forward voltammetric scans (see Fig. 3c). Based on these data, the electron transfer kinetics are in the order ML>BL>multilayer.

In Fig. 3d, there is a clear correlation between the $E_{1/2}$ map and the Raman map of the same area (Supplementary Fig. 2g). The trend in $E_{1/2}$ is: $-0.316 \pm 0.018$ V (ML, region A) > $-0.378 \pm 0.017$ V (BL, region B) > $-0.409 \pm 0.025$ V (multilayer, region C) graphene derived from a histogram of $E_{1/2}$ values (Fig. 3e). The $|E_{3/4} - E_{1/4}|$ map (Supplementary Fig. 5c) also shows a strong correlation with the number of graphene layers,

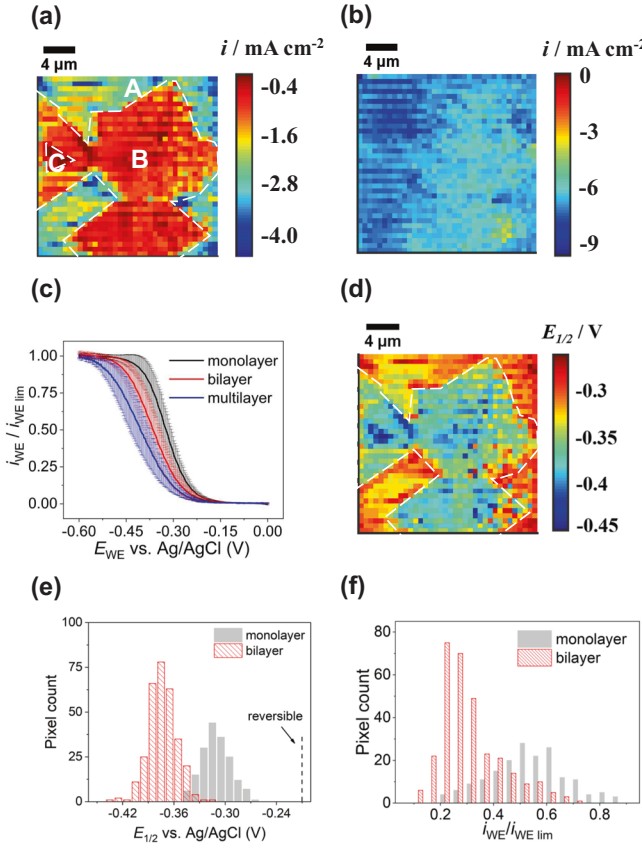

**Fig. 3 SECCM ET rate measurements of area 2.** SECCM images of current density for [Ru(NH₃)₆]³⁺ reduction at **a** −0.33 V and **b** −0.60 V vs. Ag/AgCl quasi-reference counter electrode, extracted from Supplementary Movie 2. The corresponding co-located field emission scanning electrode microscopy and Raman images and analysis of this area are given in Supplementary Fig. 10b and 2f–h, respectively. **c** Averaged linear sweep voltammetry for monolayer graphene (region A) from 176 measurements, bilayer graphene (region B) from 308 measurements and multilayer graphene (region C) from 10 measurements with y error bars. The scan rate was 0.5 V/s. **d** $E_{1/2}$ map of area 2. **e** $E_{1/2}$ values on monolayer graphene ($N = 176$) and bilayer graphene ($N = 308$) of area 2. **f** Histogram of $i_{WE}/i_{WE,lim}$ values at −0.33 V for monolayer ($N = 176$) and bilayer graphene ($N = 308$).

indicative of the identified trend in ET kinetics (see also Supplementary Fig. 5d).

Analysis of the $i_{WE} − E_{WE}$ curves at each pixel enabled the deduction of $i_{WE}/i_{WE,\ lim}$ at −0.33 V, where we make the ET kinetic analysis (Supplementary Note 6). First, with the approach outlined in Supplementary Note 6, which makes no assumption as to the ET mechanism, but assumes steady-state conditions and a uniformly accessible electrode, we use Eq. (6) in Supplementary Note 6, to deduce $k_{ML}/k_{BL} = 4.22 ± 1.42$. The data in area 1 yield a similar ratio $k_{ML}/k_{BL} = 4.37 ± 2.52$. We compare these measured kinetic ratios to the predictions from DFT calculations and rate theory below. Further, since ET kinetics often prescribe a Butler–Volmer model, we also developed a finite element method (FEM) numerical model (Supplementary Note 7), from which we deduce standard rate constants, $k_0$ and transfer coefficient, α (Supplementary Table 3) and use the data to simulate the apparent waveform with these parameters from which we can also deduce $k_{ML}/k_{BL}$ (3.3 for area 1 and 2.8 for area 2), providing confidence in the kinetic assignment.

While the ratio of ML to BL kinetic ratios in areas 1 and 2 is similar, there is a difference in the baseline ML kinetics in the 2

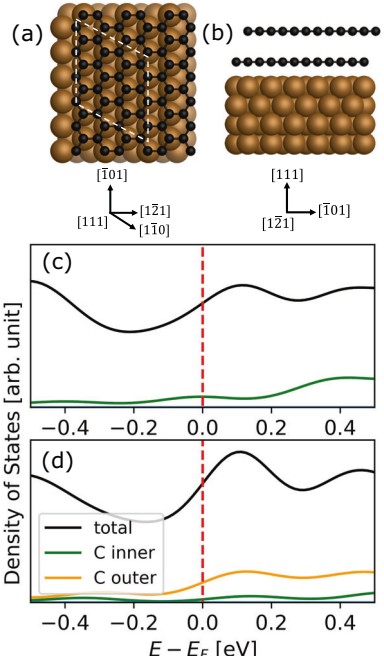

**Fig. 4 Density of states of monolayer and bilayer graphene on Cu(111).** **a** Top and **b** side view of graphene adsorbed on Cu(111). Carbon and copper atoms are shown in black and brown respectively, with the unit cell shown as a dotted white line. Density of states (DOS) centered around the Fermi level ($E_F$) for monolayer **c** and bilayer **d** graphene on Cu(111) calculated at a fixed external potential of −0.33 V vs. Ag/AgCl quasi-reference counter electrode. Shown are the total DOS and the projected DOS for graphene. The red dashed line corresponds to $E_F$. Structures were rendered using PyMOL.

areas. EBSD measurements of the copper surface were performed on areas 1 and 2 to elucidate the underlying crystallography of the support (Supplementary Note 9). The orientation of area 1 was close to Cu(111), whereas area 2 was mainly close to Cu(100). In the absence of oxygen, the electronic structure of graphene/Cu close to the Fermi level is essentially identical for Cu(100) and Cu(111)[93], but is different with oxygen present due to oxygen intercalation between Cu(100) and graphene, forming a (√2 × 2√2)R45° superstructure, which does not occur for Cu(111). As a consequence, and because graphene preferentially aligns with the Cu(111) surface, we focused exclusively on this case for atomistic modeling (*vide infra*)[93].

**Density Functional Theory prediction of the electronic structure of graphene/copper electrode.** We seek to understand the difference in OS-ET rates of ML and BL graphene on copper, and as a model system undertake DFT calculations of graphene layers adsorbed on a single-crystal Cu(111) surface (see Fig. 4a and b). In this arrangement, the graphene lattice constant, as predicted by the well-tested dispersion-inclusive PBE + vdW^surf exchange correlation functional,[94,95] is ~4% strained compared to a free-standing graphene layer (2.57 Å vs. 2.46 Å). The complete computational details of our calculations are summarized in Supplementary Note 10.

When comparing the DOS of the copper electrode functionalized with ML and BL graphene, we find that, in both cases, the graphene states do not significantly contribute to the DOS at the Fermi level. This is true when exposed against vacuum (see Supplementary Fig. 12a) as well as when applying a fixed electrochemical potential of −0.33 V vs. Ag/AgCl QRCE 56 mV vs. SCE; see Fig. 4c and d. This potential was chosen as it is in the kinetic region for all experimental voltammograms and

approximates to the halfwave potential region of the ML system in the experiments. As can be seen from the band structures of graphene-functionalized Cu(111) (see Supplementary Fig. 12b–d), graphene states are only weakly hybridized with the copper states, and adsorption is dominated by long-range dispersion interactions. This suggests that, during a reduction reaction, electrons are most likely transferred from the Fermi level dominated by metal states rather than from states localized at the graphene layers. This conclusion is different from Ni-graphene electrodes where strong hybridization between Ni and graphene takes place and leads to transfer from the first graphene layer rather than Ni[96]. In the case of graphene on copper, we find almost negligible DOS contribution of ML graphene at the Fermi level and only little contribution of the outer graphene layer in BL graphene, which likely is not strongly coupled to the metal. We therefore find it highly unlikely that a similar conclusion as for Ni can be reached in the case of copper and we prefer the conclusions that electrons are transferred from copper.

Whereas DFT calculations show that graphene adsorption does not dramatically change the electronic structure of the electrode at the Fermi level, adsorption of subsequent layers of graphene has a significant effect on the electrostatic potential drop above the electrode. As shown in Fig. 5b, the adsorption of a single layer of graphene significantly reduces the work function of the metal in vacuum ($W_e^M$) and the work function of the metal in water at zero bias potential ($W_e^{M/S}$). As schematically shown in Fig. 5a, these quantities correspond to the work (in eV) required to move an electron from the metal to vacuum ($W_e^M$) or from the metal via the solvent to vacuum ($W_e^{M/S}$). Adsorption of a second graphene layer increases $W_e^M$ and $W_e^{M/S}$ again. As shown in Fig. 5a, both of these quantities are connected via the contact potential (or Volta potential), $_S\Delta_M\Psi$ ($e_0{}_S\Delta_M\Psi$ is the work associated with the Volta potential), defined as[97]

$$_S\Delta_M\Psi = \frac{1}{e_0}(W_e^{M/S} - W_e^M) \qquad (5)$$

To assess the ability of DFT to accurately predict the electrode potentials, we compare the computed and independently measured Volta (or contact) potential difference of a typical graphene on copper substrate, and the results are given in Fig. 5c. Our calculations find that BL graphene on Cu(111) has a contact potential that is 210 mV higher than that of ML graphene on Cu(111). To connect these calculations on an idealized graphene/Cu(111) electrode with the SECCM measurements conducted on graphene grown on Cu foil, we performed Kelvin probe force microscopy (KPFM) measurements on a typical electrode. The surface potentials measured on the same graphene sample as for the SECCM data (Supplementary Note 14) shows that the BL region has a surface potential that is 100 mV higher than the ML region. This qualitative agreement between KPFM measurements and our calculations makes us confident that the computed electrostatics and potentials are a real feature of the studied electrodes.

The differences in $W_e^{M/S}$ and contact potential between ML and BL graphene translate into a different behavior of the electrodes at realistic electrochemical conditions of electron transfer, i.e. a bias voltage of −0.33 V vs. Ag/AgCl QRCE; the ML graphene electrode is more negatively charged than the BL electrode. As a consequence, the electrostatic potential of the electrode felt by the redox couple decays more gradually into the solvent for ML graphene than it does for BL graphene (see Fig. 5d).

**Correlating ET kinetics to variation in surface potential between graphene layers**. Next, we combine the insight gained from constant potential DFT data with the electrochemical rate

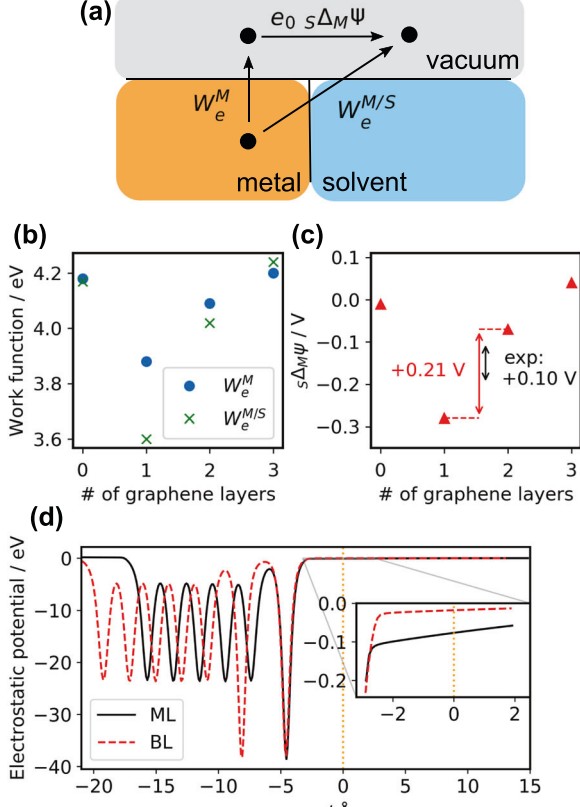

**Fig. 5 Work function, contact potential and electrostatic potential of monolayer and bilayer graphene on Cu(111). a** Scheme that defines work function of the metal when exposed to vacuum $W_e^M$, work function when exposed to solvent (work associated with the potential of zero charge) $W_e^{M/S}$, and the Volta potential (contact potential) $_S\Delta_M\Psi$. Adapted from ref. [97]. **b** $W_e^M$ (blue circles) and $W_e^{M/S}$ (green crosses) as a function of number of graphene layers. **c** Contact potential, $_S\Delta_M\Psi$, as a function of graphene layers compared against the monolayer/bilayer contact potential difference measured by Kelvin probe force microscopy (KPFM) (red: calculation, black: KPFM measurement). **d** Electrostatic potential of monolayer (ML) and bilayer (BL) graphene on Cu(111) in the direction perpendicular to the electrode, calculated at a fixed external potential of −0.33 V vs. Ag/AgCl quasi-reference counter electrode. The dashed orange line denotes the position of the Ru atom in [Ru(NH₃)₆]³⁺ and corresponds to the height, $d$, that enters the model Hamiltonian.

theory and the SNA model Hamiltonian to understand the nature of the ET and why the ET rate changes as a function of the number of graphene layers. The use of SNA and general electrochemical rate theory enables us to consider both adiabatic and non-adiabatic OS-ET kinetics, and to discriminate between these two mechanisms. In the strong molecule–metal coupling regime, the reaction is adiabatic and the difference in rates is exclusively due to the changes in the activation energy $\Delta\Delta G^\ddagger_{adiabatic}$. As derived in the Supplementary Note 13, $\Delta\Delta G^\ddagger_{adiabatic}$ for ML and BL graphene on Cu(111) can be approximated from the SNA Hamiltonian as:

$$\Delta\Delta G^\ddagger_{adiabatic} = \left[\epsilon'^{\ddagger}_{ML} - \epsilon'^{\ddagger}_{BL}\right] \cdot \bar{n}, \qquad (6)$$

where $\bar{n}$ is the occupation number of the redox state on the molecule and $\epsilon'^{\ddagger}_{ML}$ and $\epsilon'^{\ddagger}_{BL}$ are the energy levels of the respective redox states at the transition state for the ML and BL electrodes.

These energies depend on the fixed electrode potential, $E$, and electrostatic potential, $\phi$, at distance, $d$, from the electrode:

$$\epsilon'(d, E) = \epsilon(d) - 2\lambda(z - n) - \phi(d) + eE \qquad (7)$$

where $\lambda$ is the reorganization energy and $z = 3$ is the charge of the redox center in the initial state. In a step-by-step derivation laid out in the Supplementary Note 13, we show that, for this system, the activation energy reduces to a difference between the ML and BL electrostatic potential. This is because the electronic structure for the ML and BL electrodes are not significantly different (Fig. 4 and Supplementary Fig. 14). Also the redox energy $\epsilon(d)$ remains unchanged for ML and BL systems. The redox orbital on $[Ru(NH_3)_6]^{3+/2+}$ is half-filled for both systems at the transition state and hence $\bar{n} = 0.5$ in both cases. Because $[Ru(NH_3)_6]^{3+/2+}$ is an outer-sphere redox couple, we can assume that the solvent rearrangement contributions to Eq. (7) are similar for the ML and BL systems. Also, the comparison between both systems is made at the same electrode potential, $E$. This leaves the electrostatic potential at the electrode under fixed bias voltage $\phi$ as the main contribution to the change in barrier:

$$\Delta\Delta G^{\ddagger}_{adiabatic}(d) = [-\phi_{ML}(d) - (-\phi_{BL}(d))] \cdot 0.5 \qquad (8)$$

Whereas DFT calculations show that graphene adsorption on copper does not dramatically change the electronic structure of the electrode at the Fermi level, constant potential DFT calculations show that the number of graphene layers has a significant effect on the electrostatic potential drop above the electrode at the same electrode potential, as shown in Fig. 5d. From structural optimizations, we find that the equilibrium distance between the Ru atom in the redox molecule and the outer graphene layer is 4.5 Å, the electrostatic potential energy difference from constant potential DFT is about 0.07 eV. According to Eqs. (8) and (6) this translates to a difference of approximately 4 in the adiabatic rate constants, $k_{ML}/k_{BL}$, at the equilibrium distance which is in excellent agreement with the experimental measurements reported above. Displacing the Ru-center from the equilibrium position by 1.0 Å closer to (further from) the electrode results in the rate ratio of 7.5 (2.7) which shows that the predicted values for the adiabatic case are robust and close to the experimentally measured ratios regardless of uncertainties that may arise from the standard approximations inherent in our DFT treatment.

The above analysis highlights the importance of the electrostatic potential in modulating adiabatic OS-ET within the double layer. This conclusion is supported by the studies of Hromadová and Fawcett[98] which demonstrated that the potential of zero charge and crystal facet dependent work function played a decisive role in $Co[NH_3]_6^{3+}$ reduction on a range of Au single crystal electrodes in 50 mM KCl electrolyte solution. An opposite conclusion was reached by Iwasita et al. for strong electrolyte solutions (1 M KF) for a range of electrode materials[15]. We also note that electrostatic potential profile obtained with the Solvated Jellium Model (SJM) model is in good agreement with more refined modified Poisson-Boltzmann models[99–101] which faithfully model the double-layer capacitance and electrostatic potential in 100 mM electrolytes until the surface charge density is above 0.04 e/Å² (in our work the surface charge is below ~0.01 e/Å²). Another possible complication arises from the presence of $Cl^-$ in the electrolyte and possible ion-pairing, which can affect the ET kinetics. This can be addressed using the Fuoss model[102] of ion-pairing, which was adopted by Brown and Sutin to address ion-paring effects in the exchange ET of Ru-complexes (incl. $Ru[NH_3]_6^{3+}$[103]). Within this semi-quantitative model, the ion-pair formation affects both the ET rate prefactor and the barrier. The former is constant for a given redox couple-electrolyte system affecting only absolute values and is cancelled when considering rate ratios as done

herein. The formation of ion pairs modifies the reaction barrier through electrostatic interactions and within the SNA framework the *absolute* rate constants would change by a factor of 2 when going from 0 mM to 100 mM electrolyte. However, the contribution from ion pairs cancels out when computing the ratio of rates at ML and BL graphene and does not affect our analysis. Hence, our computational model is expected to accurately capture the electrostatic potential double-layer and model the experiments carried out in 50 mM KCl.

To determine the adiabaticity of the OS-ET, the prefactor was addressed next (see Supplementary Note 13 for details). The distance between the $[Ru(NH_3)_6]^{3+/2+}$ and the metal differs by about 3 Å between ML and BL graphene[104]. If the ET had a significant non-adiabatic contribution, we would expect this to affect the kinetics and be reflected in the distance-dependent coupling constant, $V(d)$, entering the pre-exponential factor in Eq. (1):

$$\kappa_{nonadiabatic} = \frac{2\pi}{h} \frac{V(d)^2}{\sqrt{4k_B T \lambda}} \qquad (9)$$

As described in Supplementary Note 13, both the coupling term, $V$, and the reaction barrier are treated in detail to analyze non-adiabaticity. The analysis shows that the rate ratio is very sensitive to the distance between the redox center and the electrode, and even small displacements from the equilibrium position result in rate ratios inconsistent with the experimental observations. Within our general model, we also find that only a very weak distance-dependence in the coupling constant would enable us to rationalise the experimental difference in ET rate between ML and BL graphene (see Supplementary Note 13). These results strongly suggest that the ET reaction is adiabatic or, at best, very weakly non-adiabatic. From this rate analysis, we can conclude that the graphene layers merely modify the electrostatic potential perceived by the $[Ru(NH_3)_6]^{3+/2+}$, whereas the ET is still adiabatically conducted between the redox couple and the underlying Cu metal support. This conclusion could only be reached with the integration of the experimental design (where the effect of graphene on ET at the Cu electrode is assessed side-by-side at ML and BL motifs), microscopic SNA model Hamiltonian and rate theory, and (constant potential) DFT simulations.

Knowledge of how the structure and composition of an electrode affects the mechanism and kinetics of electrochemical reactions is of vital importance in the design of electrochemical devices with applications spanning energy storage, electrochemical sensors, electrocatalysis, and biochemical analysis. In this paper we outlined that even for the simple case of OS-ET, it has proven challenging to answer the basic question of adiabatic vs. non-adiabatic control in past work. Taking graphene as-grown on copper as a model 2D material/metal-supported electrode, we have shown that the combination of voltammetric SECCM, with complementary co-located microscopy techniques, microscopic theory, and DFT calculations, constitutes a powerful approach to determine mechanistic details that govern ET kinetics of the $[Ru(NH_3)_6]^{3+/2+}$ couple.

CVD grown graphene on Cu foil produces a nanostructured electrode that exposes monolayer, bilayer, and multilayer graphene domains. These can be addressed readily and unambiguously with SECCM, which is able to target particular features on an electrode surface and deliver high mass transport rates. Our measurements revealed a trend in kinetics from monolayer (fastest), bilayer and multilayer (slowest) graphene on copper. This trend is opposite to what has been found for graphene on Si/SiO₂[37,39]. This emphasises that the nature of the graphene support has a profound effect on OS-ET kinetics, as is

also evident in studies of OS-ET at bilayer graphene on gold vs. Si/SiO₂[30].

To rationalize our findings, we have put forward an extended theoretical model of OS-ET, which describes non-adiabatic and adiabatic regimes. Using constant potential DFT, we have established and validated an atomistic model of the electrode/ electrolyte interface. By parameterizing our extended SNA theoretical model Hamiltonian with DFT data and combining this with rate theory, we reproduce the experimental result of faster ET at monolayer compared to bilayer graphene and we establish that the reaction proceeds in a predominantly adiabatic ET regime. Varying the number of graphene layers modifies the electrostatic potential felt by the redox couple, which, in turn, changes the activation barrier for OS-ET.

Our analysis shows the strength of complementary theoretical, computational, and experimental analysis in modern electro-chemistry that we believe will be applicable and powerful for a large body of electrochemical applications of nano-functionalized electrode materials.

## Methods

**Chemicals and materials**. All aqueous solutions were prepared from ultrapure water (SELECT-HP, Purity, 18.2 MΩ cm resistivity at 25 °C). Potassium chloride (KCl, ACS grade) and hexaamineruthenium (III) chloride ([Ru(NH₃)₆]Cl₃, purity 99.9%) were purchased from Sigma Aldrich. Fresh solutions comprising of 1 mM [Ru(NH₃)₆]Cl₃ in 50 mM KCl electrolyte were prepared prior to each experiment. Silver-chloride coated silver (Ag/AgCl) wires were used as quasi-reference counter electrodes (QRCEs) for SECCM[88]. All potentials are reported against the QRCE in the solution defined (56 mV vs. saturated calomel reference (SCE)), referred to herein as Ag/AgCl. High quality n-type silicon/silicon dioxide substrates (Si/SiO₂, 525 μm thickness with 300 nm of thermally grown SiO₂) were obtained from IDB Technologies Ltd. Copper foil was purchased from Alfa-Aesar (purity 99.8%, 0.025 mm thick, product number 46365).

**Preparation of CVD graphene**. Graphene substrates were grown on polycrystal-line copper foils as reported previously[105]. Copper foil was pre-treated by elec-trochemical polishing[106], then placed in a 1″ diameter tube furnace that was pumped to 4.4 mTorr by a turbomolecular pump. The foil was heated to 1000 °C and annealed for 10 min under 5 standard cubic centimeters (sccm) hydrogen, followed by exposure to a gas mixture of 10 sccm hydrogen and 3 sccm methane (purity 99.95%) for a growth time of 25 min. After cooling under hydrogen to room temperature (over a period ca. 1 h), the graphene covered copper foil was removed from the furnace and stored. Before study, the rear (Cu) surface of the Cu/graphene sample was fixed to a gold (300 nm) layer evaporated on an Si/SiO₂ wafer, using silver paint (Agar Scientific, Ltd, U.K.). The sample was then connected to a Cu wire, ready for electrochemical measurements.

Depending on the electrode reaction, consideration needs to be given to the effect of atmospheric contamination or exposure to ambient conditions, of the graphene sample[17,37,39,68,89,107]. An important point to bear in mind for SECCM studies of graphene[37,39], and also for other local electrochemical measurements[24,30], is that because all of the different structural motifs are assessed on the same sample, they all have the same history. The [Ru(NH₃)₆]³⁺/²⁺ process displays fast kinetics (close to reversible on the SECCM timescale) on carbon electrode materials even after extensive exposure to ambient atmosphere[68,107–109], and so we can be confident in the ratio of ET kinetics measured at ML and BL graphene. Nonetheless, even if surface contamination were to provide some charge transfer resistance, it would have a similar effect on ML and BL domains and so the effect would be to make the ratio $k_{ML}/k_{BL}$ (slightly) larger, which does not affect the conclusions and analysis (Supplementary Note 13).

**Pipet fabrication**. Pipets used for SECCM were pulled from borosilicate theta capillaries (TG C150-10, Harvard Part No. 30-0114) using a Sutter P-2000 laser puller (Sutter Instruments, USA). The total inner diameters of pipets were in the range 600 nm–1 μm, determined accurately by scanning electron microscopy (SEM) using a Zeiss SUPRA 55FE-SEM. The outer walls of the pipets were ren-dered hydrophobic by silanization with dichlorodimethylsilane (99 + % purity, Acros), by flowing argon through the pipet to protect the inside from silanization.

**Voltammetric SECCM setup**. The setup and instrumentation for SECCM has been reported in previous papers from our group[89,110]. The Warwick Electro-chemical Scanned Probe Microscope platform used herein centered on a bespoke LabVIEW program controlling a National Instruments Field Programmable Gate Array (FPGA) card (model PCIe-7852R), to output voltage signals to synchro-nously control the Physik Instrument *xyz* piezoelectric positioning system and the SECCM electrochemical cell, and acquire the resulting data (tip position and

current-related signals). Referring to Fig. 1, a bias voltage $V_1$ (200 mV) was applied between two Ag/AgCl QRCEs inserted into the tapered pipet filled with 1 mM [Ru(NH₃)₆]³⁺ as the chloride salt, in 50 mM KCl electrolyte, to generate an ion conductance current ($i_{DC}$). The z-position of the pipet (normal to the substrate surface) was modulated sinusoidally (268 Hz, 40 nm peak amplitude) by a lock-in amplifier (Stanford Research, SR830), generating an alternating current component in the ion conductance current at the same frequency ($i_{AC}$). $i_{AC}$ is negligible when the pipet is in air, but has a measurable amplitude when the pipet is in meniscus contact with the surface[89,110]. A threshold (setpoint) value of $i_{AC}$ magnitude (6 pA) was set to position the meniscus on the sample. The effective potential of the sample (working electrode) with respect to the QRCEs ($E_{WE}$) is $-(V_2 + V_1/2)$, which could be changed via the value of $V_2$ while maintaining the QRCE bias voltage, $V_1$[56]. The electrochemical current (density) flowing through the sample is denoted as $i_{WE}$.

SECCM was implemented in a voltammetric hopping mode[59]. The pipet was approached towards the surface at a speed of 0.3 μm s⁻¹, and once the meniscus had contacted the surface, the pipet was held in that position for 0.1 s before a potential sweep (0.5 V s⁻¹) was applied, from the starting potential of 0 V to –0.6 V and back to the starting potential to generate CVs for [Ru(NH₃)₆]³⁺ reduction. The pipet was then retracted a distance of 1.5 μm at a speed of 2 μm s⁻¹ to break the meniscus contact with the working electrode surface and the pipet was moved to the next pixel with a hopping distance of 600 nm in the *x-y* plane at a rate of 1 μm s⁻¹. This protocol was repeated at each pixel in the area of interest.

The potential waveform output from the FPGA had a resolution of 5 μV and electrochemical current-potential data were recorded every 3 mV. Potentiodynamic movies, containing 121 frames, were plotted every 10 mV. All data analyses were performed with Matlab (R2014b, Mathworks).

Current maps herein are presented as current density to allow comparison between tips of different sizes. Meniscus landing did not leave sufficiently clear footprints to allow SECCM electrode areas to be measured directly, and so to estimate the current density we used the tip size, which is a good approximation of the contact area for SECCM measurements on graphene, graphite, and metal and metal oxide surfaces[37,39,48,56,111]. Maps of the DC ion conductance current (between the 2 QRCEs across the meniscus), obtained during SECCM measurements, reveal information on the meniscus stability and shape[39,56,108]. DC ion conductance current histograms obtained from SECCM scanning of area 1 at −0.1 V (foot of the wave) and −0.6 V (diffusion-limit region) are detailed in Supplementary Note 15. The distributions are similar for monolayer and bilayer regions, indicating that meniscus wetting is relatively insensitive to the number of graphene layers, and the trends that we see in the current maps are due to intrinsic kinetics linked to the number of graphene layers. SECCM generates reasonably fast steady-state mass transport conditions at the scan rates employed and the voltammetric waveshape is sigmoidal (vide infra)[56].

**Graphene structural characterization**. *FE-SEM*. The morphology of the scanned area (particularly to identify graphene grains) was imaged on a Zeiss Supra 55-VP FE-SEM by using the in-lens secondary electron detector, operated at 10 kV.

*Raman microscopy*. Raman spectra of graphene were obtained with a Renishaw Invia micro-Raman spectrometer, using a diode-pumped solid-state laser (Renishaw RL523C50) with an excitation wavelength of 532 nm. Spectra were obtained at 100% power with an integration time of 10 s. The same parameters were used for Raman spectroscopy mapping of graphene/Cu samples, with a 50× lens and a step size of 1 μm. For imaging purposes, the peak intensities of the 2D peak and G peak were extracted for each pixel to provide the 2D/G ratio that was plotted as maps (vide infra). The full width half-maximum (FWHM) values (2D) of ML and BL graphene for the 2 areas were analysed by choosing 60 pixels randomly in each graphene region and the corresponding histograms were plotted.

*Electron backscatter diffraction (EBSD)*. The crystal orientation of the Cu foil surface underlying the graphene areas imaged by SECCM was determined by FE-SEM EBSD (Zeiss Sigma) imaging with a Nordlys F (Oxford Instruments) camera. Measurements were performed at an accelerating voltage of 20 kV, with a collec-tion step of 0.4 μm, with the sample tilted at 70°. Data analysis was performed using Aztec 3.1 (Oxford Instruments).

**Density functional theory calculations**. Density functional theory (DFT) calcu-lations were performed with the all-electron atomic-orbital code FHI-aims using the default *tight* basis set definition[112] and the GPAW code[113–115] using the projector-augmented wave method[116] (local double-zeta polarized basis[117] and a grid spacing of 0.2 Å). We modeled the graphene adsorption on a Cu(111) surface using a dispersion-inclusive Perdew-Burke-Ernzerhof (PBE) functional that cor-rectly accounts for screening effects in the metal substrate (PBE + vdW^surf)[95]. Slab models for a clean Cu(111) surface, and for 1, 2, or 3 layers of graphene on Cu(111) were relaxed in vacuum and under electrochemical conditions using a fixed potential, grand-canonical DFT formalism[99] using the solvated jellium approach[118] and a continuum solvation model for water[119]. Density of states (DOS) and other properties were calculated for the different electrodes in vacuum, against a potential of zero charge, and at a fixed bias potential relevant to the experimental

measurements. To ensure the choice of unit cell did not affect the results, quantities such as the DOS, contact potential and electrostatic potential decay were recalculated using a different unit cell to establish convergence. Further computational details can be found in Supplementary Notes 10, 12 and 13.

## Data availability
The datasets generated during and/or analyzed during the current study are available from the corresponding authors upon reasonable request.

## Code availability
All employed code is either available via released software or from the corresponding authors upon reasonable request.

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

## Acknowledgements

D-Q.L. thanks the China Scholarship Council-University of Warwick joint scholarship programme. S.C. acknowledges funding by the EPSRC Centre for Doctoral Training in Diamond Science and Technology (EP/L015315/1). R.J.M. acknowledges funding via a UKRI Future Leaders Fellowship (MR/S016023/1) and computing resources provided by the Scientific Computing Research Technology Platform of the University of Warwick, the EPSRC-funded HPC Midlands Plus Centre for high-performance computing (EP/P020232/1) and the ARCHER2 UK National Supercomputing Service (https://www.archer2.ac.uk) via the EPSRC-funded High End Computing Materials Chemistry Consortium (EP/R029431/1). M.M.M. acknowledges funding by the Academy of Finland (projects 307853 and 317739) and the computational resources provided by CSC—IT Center for Science, Espoo, Finland (https://www.csc.fi/en/). P.R.U. thanks the Royal Society for a Wolfson Research Merit Award. M.K. acknowledges support from the Leverhulme Trust for an Early Career Fellowship. The authors thank Ashley Page for writing MATLAB programs and support, and Dr. Cameron Bentley and Dr. Sze-yin Tan for helpful discussions.

## Author contributions

D.-Q.L.: Conceptualization, Formal Analysis, Investigation and Validation, Visualization, Writing–original draft, review & editing; M.K.: Conceptualization, Formal Analysis, Visualization, Software, Writing–review & editing; D.P.: Formal Analysis, Visualization, Software; C.-H.C.: Supervision, Investigation, Formal Analysis; G.W.: Investigation, Formal Analysis; X.X.: Investigation; S.C.: Investigation, Formal Analysis, Visualization; Z.P.L.L.: Investigation; N.R.W.: Investigation, Formal Analysis, Supervision, Writing–review & editing; G.N.M.: Investigation, Formal Analysis, Visualization; M.M.M.: Conceptualization, Formal Analysis, Investigation and Validation, Visualization, Writing–original draft, review & editing; R.J.M.: Conceptualization, Formal Analysis, Investigation and Validation, Visualization, Writing–original draft, review & editing; P.R.U.: Conceptualization, Formal Analysis, Supervision, Writing–original draft, review & editing.

## Competing interests

P.R.U. is a co-author with N. Ebejer of granted patent PCT/GB2011/051518 "Pipets containing Electrolyte and Electrodes", which describes dual-channel SECCM. The remaining authors declare no competing interests.
