## [Peer Review File · Nature Communications]

Adiabatic versus non-adiabatic electron transfer at 2D electrode materialsEditorial Note: Parts of this Peer Review File have been redacted as indicated to remove third-party material where no permission to publish could be obtained.

REVIEWER COMMENTS

Reviewer #1 (Remarks to the Author):

Report on the manuscript:

Adiabatic versus Non-Adiabatic Electron Transfer at 2D Electrode Materials
by Dan-Qing Liu et al.

The authors report on a combined experimental and theoretical investigations of an outer-sphere electron transfer reaction on graphene layers supported on copper. The experimental part uses an elegant combination of various techniques which allows a characterization of the substrate on a nanoscopic scale. However, there is a related work on electron transfer, though not in an electrochemical environment, by Ravikumar et al. (DOI: 10.1039/c7nr08737c) with similar experimental results, which has not been cited by the authors.

While the experimental results are convincing, we have several issues with the theoretical part.

- Starting from the abstract, where the authors write: microscopic understanding is largely based on idealized theories, developed in isolation from experiments...., and similar statements in the introduction. By definition, all theories are idealized; the theory developed in ref. 80, on which the authors base much of their explanation, was instigated by the experimental works of ref. 15, in which the author of ref. 80 participated.

In contrast to what the authors write, the principal parameters of the theory of outer-sphere electron transfer reaction are well defined: energy of reorganization, electronic coupling, and solvent dynamics. In principle, they can be measured and calculated, though this is not an easy task. Indeed, on p. 11 the authors write: As all quantities entering the SNA Hamiltonian can be obtained from rst principles calculations, - this is not true for the energy of reorganization, which would require very heavy computing. However, results obtained from classical simulations appear reliable.

- The authors claim that they have extended the theory of ref. 80 by adding a term describing the influence of the electrostatic potential. But this term is trivial and already implicit in ref. 80.

- In essence, the authors observed that the electron transfer on a monolayer of graphene is about four times faster than on a bilayer. When comparing the rates of an electron transfer at different metals, a difference of up to a factor of two is not unusual, and can be caused by secondary effects like an effect of the metal on the double-layer structure or dynamics, not to mention experimental errors. But a factor of four requires some explanation. To explain their results, the authors have performed two kinds of calculations:

- They have calculated the electronic densities of states of pure copper, and of copper in the presence of one or two layers of graphene (see Fig. 3). We note in passing, that the unit cell employed is rather small for an incommensurate layer. Looking at the results, we are not sure if these DOS have been properly normalized - obviously only the surface DOS is relevant and not the total DOS. In any case, the d band of copper, which figures so prominently, is irrelevant, it lies far too low to effect the electron transfer. The authors should focus on the energy range close to the Fermi level. Nevertheless the authors conclude that the electrons are exchanged directly with copper, and not with graphene. In contrast, Ravikumar et al. cited above observe a modification of the graphene DOS by the underlying nickel and discuss their results in terms of an exchange with graphene. Given the large distance between copper and the reactant, we prefer the latter interpretation.

- In addition, the authors calculate the work function of graphene-covered copper in an electrochemical environment, and base their explanations on these results. Experiments on metals have shown no effect of the work function on electron transfer rates, even though they may differ by as much as 1 eV. The differences in the work function are cancelled by the potential drop in the double layer. Anyway, the authors observe that the work function at bilayer graphene is larger by about 0.4 eV compared to a monolayer. Due to screening, this difference is diminished to

about 0.1 eV at the site of the reactant. In their calculations the electrolyte solution is modelled by a primitive though popular model with a continuum solvent and a solvated jellium. The reactant has charges of $2+/3+$, and is certainly surrounded by Cl^- ions. There is no way that a continuum model for the solution can adequately describe the double-layer in the vicinity of these ions.

In summary: The experiments are fine and can be published, though, in view of the results by Ravikumar et al., not in this journal. The theoretical explanation we find unconvincing.

Reviewer #2 (Remarks to the Author):

Recommendation: Publish with minor revision

Abstract: Using scanning electrochemical cell microscopy (SECCM), the authors evaluate the electron transfer characteristics towards the hexaammineruthenium (III/II) redox couple on heterogeneous sample of graphene grown over Cu and compare their observations as a function of number of layers (e.g. monolayer, bilayer, multilayer) with predictions from the Schmickler-Newns-Anderson model Hamiltonian. SECCM reveals that the kinetics of electron transfer on monolayer graphene on Cu are about four-fold faster than on bilayer, and that there are some few differences depending on the Cu facet explored. However, in comparison with theory, the main finding is that given the similarities in the electronic structure of the bilayer and multilayer surfaces at the Fermi level probed via DFT, the observed differences boil down to differences in the work function of the two surfaces, and thus of the electrostatic potential experienced by the redox couple at the interface. Predictions using DFT agree with the experimentally-observed fourfold difference, and strongly point to an adiabatic electron transfer case.

Comments: This is by far the best paper I have read during the whole year. It makes a such an insightful, clear, and concise exposition to the problem of comparing theory and experiment in electron transfer theory that I can't wait for this work to be published and share it with my own group, collaborators, and my electrochemistry class. Obviously, I strongly support its publication in Nature Communications. Beyond the scientific value of the discovery presented here – a systematic way to evaluate the adiabaticity of reactions on graphitic surfaces – the authors pave out a really interesting path for simulation groups to understand the context and the analysis required to explain electrochemical results. The approach will be highly impactful and highly recreated because it is also highly practical for modern simulation efforts. The electrochemistry community is badly in need of such examples.

While I feel excited about the manuscript, and would like to virtually “accept as is”, I think there are some minor aspects that deserve attention, as follows:

In the discussion in Sections S11 and S6, the definitions of some variables are missing, such as k_T for the mass transfer constant and g , for the coupling constant in Schmickler's 1986 paper. These should be explicitly mentioned and explained, if needed.

In the final comparison of experimental and theoretically-predicted kinetics in line 481, the authors mention a comparison according to Equations 6 and 8, but perhaps they mean Equation 8 and S27. I think Eq. S27 should be on the main text for clarity about how the ratio of rates was determined.

The Raman spectra in Figure S2 seem very oddly shaped, appearing more triangular-shaped rather than Lorentzian. The Cu substrate likely fluoresces significantly, so I wonder if this is an artifact of how the background was subtracted. I would suggest that in addition to the peak position and relative heights, the authors made an analysis of the peak width and compare it to existing literature to strengthen their analysis on the ML and BL nature of their samples.

The manuscript is already well referenced, so the following is just a suggestion. The authors conclude that the main driver for the electron transfer reactivity is the DOS of the underlying Cu,

and there are some supplementary experiments on Au mentioned to back this up. But they don't explore or mention the reactivity on graphene (without any metal substrate), or how they compare to the case of graphene on metal. Perhaps the following manuscript where the reactivity of Graphene over Au compared to Graphene over Si/SiO₂ on the same SECM image is useful: <https://www.sciencedirect.com/science/article/pii/S0013468616314517>.

Finally, the authors make an effort to not use the Butler-Volmer model to describe any rate constants and clearly they have good reasons which they explain. But I don't think we all have the same aversion to a model that is practical and that may allow others to reproduce the results observed here and understand them in the context of their own experiments. I suggest that the B-V rate constants from the experiments reported on Figures 2 and 3 be reported, at least in the supporting information.

Reply to the Reviewers on the manuscript entitled: "**Adiabatic versus Non-Adiabatic Electron Transfer at 2D Electrode Materials**" by Dan-Qing Liu et al.

We thank the reviewers for their valuable comments which helped to improve the manuscript. Our response to each point is given as follows: Reviewer's comments have been quoted here in black typeface and our replies in blue typeface. To identify the modifications in the manuscript, we provide a manuscript where main additions are highlighted in blue and main deletions in red-cross.

Reviewer #1 (Remarks to the Author):

Report on the manuscript: Adiabatic versus Non-Adiabatic Electron Transfer at 2D Electrode Materials by Dan-Qing Liu et al. The authors report on a combined experimental and theoretical investigations of an outer- sphere electron transfer reaction on graphene layers supported on copper. The experimental part uses an elegant combination of various techniques which allows a characterization of the substrate on a nanoscopic scale. However, there is a related work on electron transfer, though not in an electrochemical environment, by Ravikumar et al. (DOI:10.1039/c7nr08737c) with similar experimental results, which has not been cited by the authors. While the experimental results are convincing, we have several issues with the theoretical part.

We thank the reviewers for their assessment and for pointing out the work by Ravikumar et al., which we now cite and discuss in the revised manuscript.

The study of Ravikumar et al. and our study are not directly comparable (UHV vs. electrochemical environment, core-hole clock XPS vs. SECCM measurements). In our study, we conclude that the difference in electron transfer rate between monolayer and bilayer graphene on copper arises from the change in electrostatic potential decay, which we base on a detailed analysis of electronic structure and electrostatic potential under realistic electrochemical conditions. Ravikumar et al. also report that electron transfer in monolayer graphene is faster than in bilayer graphene on Nickel. They conclude that this is due to strong electronic coupling, which they base solely on an analysis of the DOS without analyzing the potential perceived by the molecule when adsorbed on monolayer or bilayer graphene. We note that the interaction of graphene and nickel is much stronger than for graphene on copper. No such strong electronic coupling is apparent or has been reported previously for graphene on copper, corroborated by our own analysis of the DOS.

We therefore do not think that the two studies are in disagreement and discuss this in the revised manuscript in the discussion section on page 19 as follows:

"This conclusion is different from the Ni-graphene electrodes where strong hybridization between Ni and graphene takes place and leads to transfer from the first graphene layer rather than Ni. [DOI: 10.1039/c7nr08737c] In the case of graphene on copper, we find almost negligible DOS contribution of monolayer graphene at the Fermi level and only little contribution of the outer graphene layer in bilayer graphene, which likely is not strongly coupled to the metal. We therefore find it highly unlikely that a similar conclusion as for Nickel can be reached in the case of Copper and we prefer the conclusions that electrons are transferred from Copper."

- Starting from the abstract, where the authors write: microscopic understanding is largely based on idealized theories, developed in isolation from experiments..., and similar statements in the introduction. By definition, all theories are idealized; the theory developed in ref. 80, on which the

authors base much of their explanation, was instigated by the experimental works of ref. 15, in which the author of ref. 80 participated.

We have revised the statements in abstract and introduction to better reflect the fact that, indeed, all theories are idealized to some extent. First principles computations face the additional challenge that direct comparison to experiment is extremely challenging and in electrochemistry, not commonly reported in literature.

In the abstract, we have revised the text as follows:

*“Outer-sphere electron transfer (OS-ET) is a cornerstone elementary electrochemical reaction, yet microscopic understanding is largely based on **idealized** theories, developed in **isolation from atomistic details or simulations**, and experiments that themselves are often close to the kinetic (diffusion) limit.”*

-In contrast to what the authors write, the principal parameters of the theory of outer-sphere electron transfer reaction are well defined: energy of reorganization, electronic coupling, and solvent dynamics. In principle, they can be measured and calculated, though this is not an easy task. Indeed, on p. 11 the authors write: As all quantities entering the SNA Hamiltonian can be obtained from first principles calculations, - this is not true for the energy of reorganization, which would require very heavy computing. However, results obtained from classical simulations appear reliable.

We fully agree that the outer-sphere parameters are well-defined. Our original statement was meant to highlight that their microscopic interpretation and calculation from atomistic simulations are difficult. We have rephrased this in the revised work.

However, we do not agree with the comment that reorganization energies would not be available with first principles methods. For instance, the constrained DFT method (<https://doi.org/10.1021/cr200148b>) has been successfully applied to compute the reorganization energy in various environments (<https://doi.org/10.1063/1.3190169>, <https://doi.org/10.1021/acs.chemmater.7b04618>). In the present work, we have not computed the reorganization energy as this is not necessary as we focus on relative electron transfer rates and the reorganization energy cancels out in the theoretical treatment.

In the 1st paragraph of the Introduction, we have revised the text as follows:

*“Much of our present theoretical understanding of electrochemical kinetics is based on rather rudimentary treatments and model Hamiltonians which describe the kinetics in terms of simple, **but physically and conceptually well-defined** parameters. **Extracting atomic level insight from these models and parameters remains challenging**, as the parameters **are often** treated as fitting parameters or obtained from first principles for simplified systems which cannot be addressed experimentally.”*

- The authors claim that they have extended the theory of ref. 80 by adding a term describing the influence of the electrostatic potential. But this term is trivial and already implicit in ref. 80.

We were unaware that the electrostatic potential is already implicitly included in the Schmickler-Newns-Anderson theory (SNA) as this is not explicitly mentioned in the original references. However, unless explicitly treated, the impact of electrostatics cannot be evaluated. To our knowledge, this term has not been explicitly considered until recently, as already cited in the original manuscript (<https://doi.org/10.1021/acs.jpcc.8b07534>, <https://doi.org/10.1021/acs.jpcc.9b03639>) and cited

additionally in the revised manuscript (<https://doi.org/10.1039/C6CP01347C>). While these works considered the electrostatic effects in electron and proton-coupled electron transfer reactions, these effects are not largely discussed in the context of SNA. **We include these terms explicitly.** The present work is the first to evaluate these terms directly from constant potential DFT and to use this information to parameterize the extended model Hamiltonian. This allows the self-consistent treatment of the electrode and to directly capture the electronic response of the electrode to variations in the potential of the electrochemical interface missing from previous treatments (<https://doi.org/10.1016/j.jelechem.2019.113664>) In the revised manuscript, we have made modifications to the text in abstract and on page 5 to better emphasize this point of novelty.

Abstract:

“To rationalize these findings we explicitly include electrostatic potential contributions to the Schmickler-Newns-Anderson model Hamiltonian for electron transfer and parametrized it using constant potential DFT.”

Bottom of Page 5:

“Here, we develop a theoretical model based on the Schmickler-Newns-Anderson (SNA) model Hamiltonian for ET accounting explicitly for the electrostatic interactions in the double layer and combine this model with constant potential density functional theory (DFT) and rate theory.”

- In essence, the authors observed that the electron transfer on a monolayer of graphene is about four times faster than on a bilayer. When comparing the rates of an electron transfer at different metals, a difference of up to a factor of two is not unusual, and can be caused by secondary effects like an effect of the metal on the double-layer structure or dynamics, not to mention experimental errors. But a factor of four requires some explanation. To explain their results, the authors have performed two kinds of calculations:

We appreciate that the reviewer agrees with us that the large difference in ET rates between monolayer and bilayer graphene (and on the same sample in the same experiment) is an important finding that deserves detailed investigation. Please note that we have also carefully assessed how the variation of certain model parameters affects this ratio and that, while the ratio varies to some extent as a function of adsorbate height, our findings robustly point towards faster kinetics on monolayer graphene.

- They have calculated the electronic densities of states of pure copper, and of copper in the presence of one or two layers of graphene (see Fig. 3). We note in passing, that the unit cell employed is rather small for an incommensurate layer.

The unit cell depicted in Figure 4b was the primitive unit cell and not the supercell used for our computations, which is a (2x2) repeat of the primitive unit cell (see blue dashed line in figure to the right). This was not made sufficiently clear in the original manuscript and has now been clarified in the computational methods section. To address the reviewers' concerns on the dependence of the results with respect to unit cell, we have performed new calculations in the unit cell shown with a green dashed line in the figure to the right.

We find that our results only weakly depend on the choice and size of unit cell. This includes the Volta (contact) potential (0.14 V vs. 0.21 V with small and large unit cell, respectively), the projected DOS of graphene at the Fermi level (no discernible difference), and the electrostatic potential decay difference between mono- and bilayer graphene (0.06 eV vs. 0.05 eV at the height of the adsorbate for the small and large unit cells, respectively). The comparison between the small and large unit cells can be found in a newly added section of the Supplemental Information (section S12), specifically in Figures S10-S13.

We have updated Figure 4 to include the correct unit cell depiction, which is reproduced further below. We have further modified the computational methods section and included a detailed description of the unit cell dependence of our results in the Supplemental Information (section S12).

- Looking at the results, we are not sure if these DOS have been properly normalized - obviously only the surface DOS is relevant and not the total DOS. In any case, the *d* band of copper, which figures so prominently, is irrelevant, it lies far too low to affect the electron transfer. The authors should focus on the energy range close to the Fermi level. Nevertheless, the authors conclude that the electrons are exchanged directly with copper, and not with graphene. In contrast, Ravikumar *et al.* cited above observe a modification of the graphene DOS by the underlying nickel and discuss their results in terms of an exchange with graphene. Given the large distance between copper and the reactant, we prefer the latter interpretation.

We thank the reviewers for their comments on the DOS analysis, which we have used to improve the relevant Figure 4c-d and the associated discussion in the revised manuscript.

Based on the paper by Ravikumar on electron transfer at Ni/graphene electrodes, we reassessed the predictions from our theoretical model. On Ni/graphene, the first graphene layer is hybridized with the underlying Ni electrode whereas the second graphene layer is only weakly hybridized. Based on these considerations, Ravikumar *et al.* concluded that the electron is transferred from the first adsorbed graphene layer rather than nickel.

In the case of graphene on copper, we find almost negligible DOS contribution of monolayer graphene at the Fermi level and only little contribution of the outer graphene layer in bilayer graphene, which likely is not strongly coupled to the metal. We therefore find it highly unlikely that a similar conclusion as for nickel can be reached in the case of copper.

From these results, we conclude that electrons are transferred from copper. In this case, there is no electronic contribution to the ratio of ET rate between monolayer and bilayer graphene. Even if we

assume that electrons transfer from the first graphene layer, the DOS contribution for the monolayer and bilayer system are identical, which also means that they cancel in the ratio between ET rates.

More importantly, we base our analysis of the adiabaticity and electrostatic potential on the distance of the $\text{Ru}[\text{NH}_3]_6^{3+}$ from the outer-most graphene layer. As the distance is always measured from the top-most graphene, and other graphene layers merely increase this distance, our results regarding adiabaticity and electrostatic potentials remain unchanged whether we consider that ET takes place from Cu or the graphene closest to Cu. Indeed, from our results we cannot certainly say whether the ET is conducted by the copper or graphene. Our results and conclusions would not be affected either way.

These points are now discussed in the revised manuscript on page 19

“This conclusion is different from the Ni-graphene electrodes where strong hybridization between Ni and graphene takes place and leads to transfer from the first graphene layer rather than Ni. [DOI: 10.1039/c7nr08737c] In the case of graphene on copper, we find almost negligible DOS contribution of monolayer graphene at the Fermi level and only little contribution of the outer graphene layer in bilayer graphene, which likely is not strongly coupled to the metal. We therefore find it highly unlikely that a similar conclusion as for Nickel can be reached in the case of Copper and we prefer the conclusions that electrons are transferred from Copper.”

The new figure 4 is reproduced below:

- In addition, the authors calculate the work function of graphene-covered copper in an electrochemical environment, and base their explanations on these results. Experiments on metals have shown no effect of the work function on electron transfer rates, even though they may differ by as much as 1 eV. The differences in the work function are cancelled by the potential drop in the double layer. Anyway, the authors observe that the work function at bilayer graphene is larger by about 0.4 eV compared to a monolayer. Due to screening, this difference is diminished to about 0.1 eV at the

site of the reactant. In their calculations the electrolyte solution is modelled by a primitive though popular model with a continuum solvent and a solvated jellium. The reactant has charges of 2+/3+, and is certainly surrounded by Cl⁻ ions. There is no way that a continuum model for the solution can adequately describe the double-layer in the vicinity of these ions.

First, we would like to note that we do not explain the variations in ET kinetics using the work function directly. Instead, we attribute the variations in ET kinetics to changes in the electrostatic potential. We connect the calculated electrostatic potentials with experimentally measured electrode properties via the contact potential.

Second, the role of the work function or double-layer effects on ET cannot be considered to be fully resolved. Studies where work function effects are found to be negligible, are performed in rather strong electrolyte solutions and high concentrations are used to minimize double-layer effects (for instance, Iwasita et al. used 1M KF (<https://doi.org/10.1002/bbpc.19850890212>). In dilute electrolytes, such as the 50mM KCl used in the present work, double layer and work functions effects seem to be more pronounced. For instance, Hromadová and Fawcett (<https://doi.org/10.1021/jp994136z>) conclusively demonstrated that the potential of zero charge and crystal facet dependent work function plays a decisive role in Co[NH₃]₆³⁺ reduction on a range of Au single crystal electrodes in 50mM KCl.

Regarding the solvated jellium model, we note that the electrostatic potential profile obtained with the solvated jellium “electrolyte” (“Jellium” in the below Figure) is very close to the electrostatic potential predicted with more refined Poisson-Boltzmann models, which are among the most accurate solvent models currently available for electronic DFT calculations. In the figure below, we show the electrostatic potential of Au(211) in 0.5 M KF obtained using constant potential DFT with a range of Poisson-Boltzmann-like models including the linearized (“Linearized”) and non-linear Poisson-Boltzmann (“GC”) models, Bikerman-Fraiese model accounting for finite-size ion effects with or without the Stern layer correction, accounting for ionic decrement to the dielectric constant (“Decrement”), an analytic Gaussian shaped counter charge (“Gaussian”), and the so-called “boundary condition” method where the simulation cell is charged but no ions are present. (<https://doi.org/10.1063/1.5047829>).

[REDACTED]

In general, the Poisson-Boltzmann predictions are expected to become worse as the electrolyte concentrations is increased, ion size is increased, at highly charged electrodes, or when the ion charge is larger than 1. In the present context, the most important quantity to produce is the electrostatic potential profile in the double layer as this is used to compute the electrostatic contributions to the rate. From the studies of Fawcett, it can be seen that for dilute 1:1 electrolytes with ionic sizes close to KCl, the electrostatic potential in the double layer is rather well reproduced by continuum dielectric and Poisson-Boltzmann models (<https://doi.org/10.1016/j.electacta.2009.02.025>, <https://doi.org/10.1063/1.1464826>, <https://doi.org/10.1021/jp9058577>) A comprehensive comparison between molecular dynamics, classical DFT, and Poisson-Boltzmann models for finite-sized ions and solvent has been made (<https://doi.org/10.1021/ct3001156>, <https://doi.org/10.1039/D1SC02289J>) showing that Poisson-Boltzmann faithfully models the double layer capacitance and electrostatic potential in 100 mM electrolytes (<https://doi.org/10.1039/D1SC02289J>) until the surface charge density is above $0.04e/\text{\AA}^2$ (reduced charge 0.5 for 3.5\AA ions). In our work the surface charge density is below $\sim 0.01e/\text{\AA}^2$. Hence, the electrostatic potential in the double-layer is expected to be sufficiently well described in our work.

Regarding the role of Cl^- and possible ion-pairing on the ET kinetics, this can be addressed using the Fuoss model (<https://doi.org/10.1021/ja01552a016>) of ion-pairing which was adopted by Brown and Sutin to address ion-pairing effects in the exchange ET of Ru-complexes (incl. $\text{Ru}[\text{NH}_3]_6^{3+}$ <https://doi.org/10.1021/ja00498a016>). Within this semi-quantitative model, the ion-pair formation affects both the ET rate prefactor and the barrier. The former is constant for a given redox couple – electrolyte system affecting only absolute values and is cancelled when considering rate ratios. The formation of ion pairs modifies the reaction electrostatic part of the reaction barrier with κr term as

$$W_r = \frac{z_1 z_2 e^2}{D_s \bar{r} (1 + \kappa \bar{r})}$$
$$\kappa = \left(\frac{8\pi N^2 e^2 \mu}{1000 D_s R T} \right)^{1/2} = \beta \sqrt{\mu}$$

where z is the charge of a reactant, r is the radius, N is the Avogadro constant, D_s is the dielectric constant and μ the electrolyte concentration. If this contribution is included in the barrier deduced from the SNA theory, the *absolute* rate constants would change by a factor of 2 in 100 mM electrolyte solutions compared to 0 mM solutions. More notably, the ion pairing effect on the relative rate constants for monolayer and bilayer graphene would cancel out completely provided the electrolyte concentration remains fixed.

Another possibility for the ion effect is to reduce the apparent charge from the nominal +3 to +1 as has been suggested in the literature (<https://doi.org/10.1021/jp054759e>, <https://doi.org/10.1016/j.electacta.2007.12.035>). While the change in the charge state will affect the electrostatic interactions in the double layer, the theoretical analysis only deals with the *changes* in the charge state $z \rightarrow z-1$ and the three stationary points of the Hamiltonian correspond to charge in initial, transition, and final states; this is why the barrier has the dependency $\phi(d)$ rather than $z \cdot \phi$ and to the free energy *change* it does not matter whether $z=+3$ or $z=+1$ as long as the change of charge corresponds to the transfer of a single electron.

We have added a discussion of these points to the manuscript on page 24:

“The above analysis highlights the importance of the electrostatic potential in modulating the adiabatic OS-ET within the double layer. This conclusion is supported by the studies of Hromadová and Fawcett (<https://doi.org/10.1021/jp994136z>) demonstrating that the potential of zero charge and crystal facet dependent work function play a decisive role in $\text{Co}[\text{NH}_3]_6^{3+}$ reduction on a range of Au single crystal electrodes in 50 mM KCl electrolyte solution. An opposite conclusion was reached by Iwasita et al for strong electrolyte solutions (1 MKF) for a range of electrode materials. We also note that electrostatic potential profile obtained with the SJM model is in good agreement with more refined modified Poisson-Boltzmann models (<https://doi.org/10.1021/ct3001156>, <https://doi.org/10.1039/D1SC02289J>, <https://doi.org/10.1063/1.5047829>) which faithfully model the double layer capacitance and electrostatic potential in 100 mM electrolytes until the surface charge density is above $0.04e/\text{\AA}^2$ (in our work the surface charge is below $\sim 0.01e/\text{\AA}^2$). Another possible complication arises from the presence of Cl^- in the electrolyte and possible ion-pairing, which can affect on the ET kinetics. This can be addressed using the Fuoss model (<https://doi.org/10.1021/ja01552a016>) of ion-pairing which was adopted by Brown and Sutin to address ion-pairing effects in the exchange ET of Ru-complexes (incl. $\text{Ru}[\text{NH}_3]_6^{3+}$ <https://doi.org/10.1021/ja00498a016>). Within this semi-quantitative model, the ion-pair formation affects both the ET rate prefactor and the barrier. The former is constant for a given redox couple – electrolyte system affecting only absolute values and is cancelled when considering rate ratios as done herein. The formation of ion pairs modifies the reaction barrier through electrostatic interactions and within the SNA framework the absolute rate constants would change by a factor of 2 when going from 0 mM to 100 mM electrolyte. However, the contribution from ion pairs cancels out when computing the ratio of rates at monolayer and bilayer graphene and does not affect our analysis. Hence, our computational model is expected to accurately capture the electrostatic potential double-layer and model the experiments carried out in 50 mM KCl.”

In summary: The experiments are fine and can be published, though, in view of the results by Ravikumar et al., not in this journal. The theoretical explanation we find unconvincing.

To address the reviewers concerns on the theoretical part of the manuscript, we summarize again all the changes we have included in the revised manuscript:

- We have adapted Figures 4 and 5 to reflect the correct unit cell and to improve the presentation of the data
- We have tested the dependence of the calculation results with respect to unit cell size and find little dependence
- We have improved formulations in the manuscript to better emphasize the importance of electrostatics and the novelty of our approach
- We have added a thorough discussion of our electrostatic double layer and solvation model, as well as an analysis of the role of chlorine ions

Reviewer #2 (Remarks to the Author):

Recommendation: Publish with minor revision

Abstract: Using scanning electrochemical cell microscopy (SECCM), the authors evaluate the electron transfer characteristics towards the hexaammineruthenium (III/II) redox couple on heterogeneous sample of graphene grown over Cu and compare their observations as a function of number of layers (e.g. monolayer, bilayer, multilayer) with predictions from the Schmickler-Newns-Anderson model Hamiltonian. SECCM reveals that the kinetics of electron transfer on monolayer graphene on Cu are about four-fold faster than on bilayer, and that there are some few differences depending on the Cu facet explored. However, in comparison with theory, the main finding is that given the similarities in the electronic structure of the bilayer and multilayer surfaces at the Fermi level probed via DFT, the observed differences boil down to differences in the work function of the two surfaces, and thus of the electrostatic potential experienced by the redox couple at the interface. Predictions using DFT agree with the experimentally-observed fourfold difference, and strongly point to an adiabatic electron transfer case.

Comments: This is by far the best paper I have read during the whole year. It makes a such an insightful, clear, and concise exposition to the problem of comparing theory and experiment in electron transfer theory that I can't wait for this work to be published and share it with my own group, collaborators, and my electrochemistry class. Obviously, I strongly support its publication in Nature Communications. Beyond the scientific value of the discovery presented here – a systematic way to evaluate the adiabaticity of reactions on graphitic surfaces – the authors pave out a really interesting path for simulation groups to understand the context and the analysis required to explain electrochemical results. The approach will be highly impactful and highly recreated because it is also highly practical for modern simulation efforts. The electrochemistry community is badly in need of such examples.

We appreciate the reviewer's positive comments and are delighted that they share our enthusiasm for the work.

While I feel excited about the manuscript, and would like to virtually "accept as is", I think there are some minor aspects that deserve attention, as follows:

In the discussion in Sections S11 and S6, the definitions of some variables are missing, such as k_T for the mass transfer constant and g , for the coupling constant in Schmickler's 1986 paper. These should be explicitly mentioned and explained, if needed.

We thank the reviewer for picking these omissions up. We have now defined k_T and k_{ET} in section S6 and the coupling strength g in Section S13 (renumbered due to the addition of 2 additional SI sections).

In the final comparison of experimental and theoretically-predicted kinetics in line 481, the authors mention a comparison according to Equations 6 and 8, but perhaps they mean Equation 8 and S27. I think Eq. S27 should be on the main text for clarity about how the ratio of rates was determined.

In relation to the reviewer’s final point about adding a Butler-Volmer analysis (new section S7) we have modified this part of the manuscript and presented ratios from the 2 methods (see below). Equation numbers have been updated. We thank the reviewer for the suggestions.

The Raman spectra in Figure S2 seem very oddly shaped, appearing more triangular-shaped rather than Lorentzian. The Cu substrate likely fluoresces significantly, so I wonder if this is an artifact of how the background was subtracted. I would suggest that in addition to the peak position and relative heights, the authors made an analysis of the peak width and compare it to existing literature to strengthen their analysis on the ML and BL nature of their samples.

We believe that the apparent “triangular-shaped” Raman peaks is visual/resolution effect. A zoom in version of the Raman peaks is shown below. The G and 2D peaks of all spectra shown in Figure S2b and S2f, have been fitted individually using a Lorentzian function. The results (Table below) shows that the peaks are fitted quite well and we believe that the background subtraction is reasonable.

Raman spectrum of monolayer of area 1 from Figure S2b with G and 2D peaks fitted with by Lorentzian function (red lines).

Table. Lorentzian fitting parameters.

	G band	2D band
Number of points	49	92
Degrees of Freedom	45	88
Reduced Chi-sqr	2.89E-5	1.73E-4
Residual Sum of Squares	1.30E-3	1.52E-2
R-Square (COD)	0.9950	0.9926
Adj. R-Square	0.9947	0.9923

In addition to the 2D/G ratio, the width of the 2D peak of graphene is an indicator of the number of graphene layers. The full width half-maximum (FWHM) values of area 1 and area 2 on monolayer and bilayer graphene are analyzed herein (see figure below). The FWHM of bilayer graphene is broadened compared with monolayer graphene and is a composite of 4 overlapping peaks (figure below). The blue shift of the 2D peak for bilayer graphene compared with the monolayer is further diagnostic of the different graphene character (<https://doi.org/10.1103/PhysRevLett.97.187401>).

The results are comparable with the Raman analysis of reported layer-by-layer stacked graphene (<https://doi.org/10.1002/adma.201003673>) and graphene on Si/SiO₂ (<https://doi.org/10.1021/acsnano.5b00550>) with an average of ~ 35 cm⁻¹ for monolayer graphene and ~ 49 cm⁻¹ for bilayer graphene. The slightly bigger FWHM (2D) values for graphene on Cu foil may can be explained by different local strain at the scale of the Raman laser spot (<https://doi.org/10.1016/j.carbon.2013.11.020>).

FWHM (2D) values from monolayer and bilayer regions of area 1 and area 2. (60 pixels chosen randomly for each region).

Representative Raman spectrum (black) and fits (red) obtained at bilayer graphene (2D peak).

As we emphasized in the paper, the Raman data were combined with co-located SEM images which give a distinct contrast variation for monolayer, bilayer and multilayer graphene (<https://doi.org/10.1007/s12274-013-0285-y>) to further confirm our assignments.

We have added discussion of these points in section S2 with the following text:

“In addition, the width of the 2D peak of graphene can be an indicator of the number of graphene layers. The full width half-maximum (FWHM) values of monolayer and bilayer graphene in area 1 were analyzed by choosing 60 pixels randomly in each graphene region. The histogram in Figure S2d indicates monolayer and bilayer graphene have average FWHM (2D) values of $\sim 35 \text{ cm}^{-1}$ and $\sim 49 \text{ cm}^{-1}$, respectively, with the latter being a composite of 4 overlapping peaks (<https://doi.org/10.1103/PhysRevLett.97.187401>, <https://doi.org/10.1021/acsnano.5b00550>). The values are comparable with the Raman analysis reported for layer-by-layer stacked graphene (<https://doi.org/10.1002/adma.201003673>) and graphene on Si/SiO₂ (<https://doi.org/10.1021/acsnano.5b00550>). The FWHM (2D) values of area 2 are consistent with area 1.”

We have expanded Figure S2 with further Raman analysis of the monolayer and bilayer regions (FWHM analysis of 2D peak on area 1 and area 2 in Figure S2d and S2h).

The manuscript is already well referenced, so the following is just a suggestion. The authors conclude that the main driver for the electron transfer reactivity is the DOS of the underlying Cu, and there are some supplementary experiments on Au mentioned to back this up. But they don't explore or mention the reactivity on graphene (without any metal substrate), or how they compare to the case of graphene on metal. Perhaps the following manuscript where the reactivity of Graphene over Au compared to Graphene over Si/SiO₂ on the same SECM image is useful: <https://www.sciencedirect.com/science/article/pii/S0013468616314517>.

We thank the reviewer for suggesting that we discuss this point. We have now addressed the issue of the metal substrate in some detail, in response to reviewer 1 and the Ravikumar paper suggested (please see reviewer 1 response). Nonetheless, we agree with the reviewer that it is worth making a broader point about metal vs. insulator supported graphene and we have done so in the Conclusions,

with the following additional text, referencing some past work, including the paper the reviewer mentions (which was already cited as ref 34 in the manuscript).

“This trend is opposite to what has been found for graphene on Si/SiO₂ (<https://doi.org/10.1021/ja3014902>, <https://doi.org/10.1021/acsnano.5b00550>). This emphasises that the nature of the graphene support has a profound effect on OS-ET kinetics, as is also evident in studies of OS-ET at bilayer graphene on gold vs. Si/SiO₂ (<https://doi.org/10.1016/j.electacta.2016.06.134>).”

Finally, the authors make an effort to not use the Butler-Volmer model to describe any rate constants and clearly they have good reasons which they explain. But I don't think we all have the same aversion to a model that is practical and that may allow others to reproduce the results observed here and understand them in the context of their own experiments. I suggest that the B-V rate constants from the experiments reported on Figures 2 and 3 be reported, at least in the supporting information.

We appreciate the reviewer's suggestion. The revised SI now has a new section (S7) where we use the finite element method to implement B-V kinetics in a simplified model of the SECCM pipette. Section S7 has a complete and detailed description of the model and how the kinetic parameters, k^0 and α , were calculated. Briefly, the model consists of a 3D representation of the pipettes used for the experiments reported in Figure 2 and 3 and the meniscus formed between the pipette and the substrate. At the substrate, the flux of Ru[NH₃]₆^{3+/2+} was defined by a potential-dependent rate following B-V kinetics (Table S1). A family of LSVs, with the same potential window and scan rate as the experiments in the main manuscript, were simulated for varying kinetic parameters at the substrate. From the series of LSVs we extracted values of $\Delta E_{1/2}$ ($|E_{1/2} - E^0|$) and $\Delta E_{1/4}$ ($|E_{3/4} - E_{1/4}|$) and built working surfaces (Figure S8) that were used to find k^0 and α (from the experimental $\Delta E_{1/2}$ and $\Delta E_{1/4}$) for the average LSV behaviors in Figures 2 and 3. With the values for the kinetic parameters we calculated the ratio of k_{ML}/k_{BL} at -0.33 V, as was done in the main manuscript, and found the values (3.3 for area 1 and 2.8 for area 2) to be comparable to the ones reported in the main text.

REVIEWERS' COMMENTS

Reviewer #1 (Remarks to the Author):

The authors have replied in great detail to the referees' comments, and have improved their manuscript substantially. Nevertheless, I am not convinced by their explanation of the observed difference in the electron transfer rate, nor by their claim to have made a significant contribution to electron transfer theory.

However, it is a well-written paper, and the scientific community should be given the opportunity to read it and examine the validity of the arguments. Therefore I propose that this article should be published in its present form.

Reviewer #2 (Remarks to the Author):

I am convinced with the replies provided by the authors to my (few and minor) concerns. I recommend publication.